# Clinical Practice in the Prevention, Diagnosis and Treatment of Vitamin D Deficiency: A Central and Eastern European Expert Consensus Statement

**DOI:** 10.3390/nu14071483

**Published:** 2022-04-02

**Authors:** Pawel Pludowski, Istvan Takacs, Mihail Boyanov, Zhanna Belaya, Camelia C. Diaconu, Tatiana Mokhort, Nadiia Zherdova, Ingvars Rasa, Juraj Payer, Stefan Pilz

**Affiliations:** 1Department of Biochemistry, Radioimmunology and Experimental Medicine, The Children’s Memorial Health Institute, 04-730 Warsaw, Poland; 2Department of Internal Medicine and Oncology, Faculty of Medicine, Semmelweis University, 1085 Budapest, Hungary; takacs.istvan@med.semmelweis-univ.hu; 3Department of Internal Medicine, Clinic of Endocrinology and Metabolism, University Hospital Alexandrovska, Medical University of Sofia, 1, G. Sofiyski Str., 1431 Sofia, Bulgaria; mihailboyanov@yahoo.com; 4The National Medical Research Center for Endocrinology, 117036 Moscow, Russia; jannabelaya@gmail.com; 5Faculty of Medicine, “Carol Davila” University of Medicine and Pharmacy, 050474 Bucharest, Romania; drcameliadiaconu@gmail.com; 6Department of Endocrinology, Belarussian State Medical University, 220116 Minsk, Belarus; tatsianamokhort@gmail.com; 7Department of Diagnostics and Treatment of Metabolic Diseases, Center for Innovation Medical Technology, The National Academy of Science of Ukraine, 01030 Kiev, Ukraine; nadejda05.1977@gmail.com; 8Centre for Continuing Education, Riga Stradins University, LV-1002 Riga, Latvia; dr.irasa@inbox.lv; 9Department of Internal Medicine, Comenius University Faculty of Medicine, University Hospital Bratislava, 826 06 Bratislava, Slovakia; payer@ru.unb.sk; 10Department of Internal Medicine, Division of Endocrinology and Diabetology, Medical University of Graz, Auenbruggerplatz 15, 8036 Graz, Austria; stefan.pilz@medunigraz.at

**Keywords:** vitamin D, recommendations, guidelines, supplementation, cholecalciferol, treatment

## Abstract

Vitamin D deficiency has a high worldwide prevalence, but actions to improve this public health problem are challenged by the heterogeneity of nutritional and clinical vitamin D guidelines, with respect to the diagnosis and treatment of vitamin D deficiency. We aimed to address this issue by providing respective recommendations for adults, developed by a European expert panel, using the Delphi method to reach consensus. Increasing the awareness of vitamin D deficiency and efforts to harmonize vitamin D guidelines should be pursued. We argue against a general screening for vitamin D deficiency but suggest 25-hydroxyvitamin D (25(OH)D) testing in certain risk groups. We recommend a vitamin D supplementation dose of 800 to 2000 international units (IU) per day for adults who want to ensure a sufficient vitamin D status. These doses are also recommended for the treatment of vitamin D deficiency, but higher vitamin D doses (e.g., 6000 IU per day) may be used for the first 4 to 12 weeks of treatment if a rapid correction of vitamin D deficiency is clinically indicated before continuing, with a maintenance dose of 800 to 2000 IU per day. Treatment success may be evaluated after at least 6 to 12 weeks in certain risk groups (e.g., patients with malabsorption syndromes) by measurement of serum 25(OH)D, with the aim to target concentrations of 30 to 50 ng/mL (75 to 125 nmol/L).

## 1. Introduction

Vitamin D is crucial for musculoskeletal health, as it plays an important role in the regulation of bone and mineral metabolism, and it can prevent and cure nutritional rickets and osteomalacia [1,2]. In addition, vitamin D receptor (VDR) expression in almost all human cells suggests, or even documents, a more widespread role of vitamin D for overall health, a notion that is supported by several experimental and epidemiological studies [1,3,4,5,6,7,8]. While there still exist knowledge gaps and controversy regarding potential extra-skeletal effects of vitamin D, there is a wide consensus that the high worldwide prevalence of vitamin D deficiency is of concern and requires actions to improve this situation [2,7,9]. Addressing this issue has to consider the unique metabolism of vitamin D, which is mainly synthesized in the skin stimulated by ultraviolet-B (UV-B) exposure, whereas nutrition is usually only a minor source of vitamin D [10]. Vitamin D from all different sources is metabolized to 25-hydroxyvitamin D (25(OH)D, calcifediol) in the liver, which is the main circulating vitamin D metabolite that is determined to assess vitamin D status. Further hydroxylation of 25(OH)D in the kidneys or certain extra-renal tissues results in the formation of 1,25-dihydroxyvitamin D (1,25(OH)2D, also called calcitriol), which exerts endocrine, autocrine, and paracrine effects as a steroid hormone [10]. Heterogeneous recommendations, regarding several issues in the practical management of vitamin D deficiency, represent a challenge for clinicians and health authorities on how to deal with this public health problem [11,12,13,14,15,16,17,18,19,20,21,22]. In this context, systematic evaluations of current vitamin D guidelines did, not only, observe a great heterogeneity of the recommendations, but it also reported a low quality score regarding the methodological processes for the majority of these vitamin D guidelines [17,18]. Table 1 and Table 2 provide an overview of selected guideline recommendations, with a focus on Central and Eastern European countries, for prevention and treatment of vitamin D deficiency, respectively.

In clinical practice, a great variability and controversy is reported regarding vitamin D testing and supplementation, thus requiring an improved guidance for clinicians [31]. Therefore, we aimed to draft an expert consensus statement covering important aspects of the clinical practice in the prevention, diagnosis, and treatment of vitamin D deficiency of adults. Rather than covering all of the above-mentioned vitamin D issues in great and extensive detail, we aimed to provide a simple, easy-to-follow guidance for clinicians. Respective data and recommendations regarding these issues in children can be found elsewhere [14,21,26,32,33,34].

## 2. Consensus Development Process

A European expert panel with 10 members and a focus on physicians from Eastern Europe, who are usually underrepresented in such groups, was established by selecting clinicians and key opinion leaders with expertise in vitamin D and related topics from their respective countries. The first and senior authors (P.P. and S.P.) of this article served as the chairs of this expert panel, who selected and invited the other panel members. The consensus-reaching process itself was conducted by using the Delphi method, which is a widely accepted tool for clinical consensus statements [35,36]. The Delphi method was applied by using SurveyMonkey^®^ for voting on various statements on a 9-point scale with the following numeric (and descriptive) anchors: 1 (strongly disagree), 3 (disagree), 5 (neutral), 7 (agree), and 9 (strongly agree). It was pre-specified that a consensus will be established, in the case that ≥75% of the participants agree with the statement, by a voting scale of 7 or above. In case of failing to reach a consensus according to this criterion, it was planned to repeat these surveys after group discussions and statement modifications, if necessary, until a consensus is reached.

After the alignment of the scope of this consensus document and recruitment of the 10 members of the expert panel, the detailed process of this work started with drafting various questions on practical issues regarding vitamin D, which were exclusively discussed and fine-tuned by the chairs. These questions were subsequently used in a first survey round in September 2021. All panel members were involved in this survey, and their answers, along with the existing literature on vitamin D, served as the basis to formulate respective statements by the chairs. At a hybrid meeting in Darmstadt, Germany on the 20 November 2021, the results of the first survey round, along with the subsequently formulated statements, were presented by the chairs. At the same meeting, each statement was subject to a group discussion and a Delphi voting round to reach consensus, which was finally successful for all statements. The 10 authors of this paper are the 10 experts who participated in the Delphi voting rounds. Each voting round was performed for the whole content of each Table or Figure, respectively. Group discussions before each voting round, however, were used to fine tune the statements according to the opinions and recommendations of the panel members. The work on this consensus document was funded and organized by Wörwag Pharma (Böblingen, Germany). The sponsor provided financial and logistic support but did not actively participate in the scientific discussions and consensus-reaching processes.

## 3. Consensus Recommendations

The final consensus statements with the level of agreement within the expert group are subsequently presented below. The evidence and considerations underpinning each statement are also outlined. It was our aim to base our work on the totality of available evidence by considering the established hierarchy of evidence levels. The expert panel group was encouraged to perform systematic literature reviews on the topics of this consensus document, but we have to acknowledge that this gathering of information did not follow a pre-specified structured process.

### 3.1. Current Situation of Vitamin D Deficiency Diagnosis, Prevention and Treatment

Epidemiological studies have documented a high prevalence of vitamin D deficiency worldwide [37]. Data from Europe showed that 25(OH)D concentrations below 20 ng/mL (50 nmol/L) and below 12 ng/mL (30 nmol/L) are observed in 40.4% and 13.0% of the general population, respectively [9]. Therefore, a huge gap exists between the recommendations of nutritional societies regarding dietary reference intakes, as well as target 25(OH)D concentrations, and the actual situation, as documented in large population surveys [38]. Public health actions are, therefore, required to improve the vitamin D status in the general population, but regional differences in vitamin D status, related to factors such as latitude, genetics, lifestyle, body composition, or dietary intake have to be considered [9,38,39,40,41]. A major issue to achieve this task is to harmonize the current heterogeneous efforts and guideline recommendations (Table 3).

**Table 3 nutrients-14-01483-t003:** Statement regarding the current situation of vitamin D deficiency diagnosis, prevention, and treatment.

Consensus Statement	Consensus Voting Scale	Level of Agreement
To ensure an adequate screening, prevention and treatment of vitamin D deficiency in the clinical practice, it is necessary to increase the awareness and improve education in the public and medical community. Moreover, national and international guidelines/recommendations should be precise regarding the risk groups that need to be screened, the adequate dosages for prevention and treatment of vitamin D deficiency, the treatment regimen as well as the follow-up to allow transfer into clinical practice.	9 (strongly agree)	80%
	8	0%
	7 (agree)	20%
	6	0%
	5 (neutral)	0%
	4	0%
	3 (disagree)	0%
	2	0%
	1 (strongly disagree)	0%
Overall agreement 100%, consensus endorsed		

### 3.2. Screening of Vitamin D Deficiency in Adults

No published study evaluated the effects of a screening program for vitamin D deficiency in the general population, so the evidence is insufficient to balance the benefits and harms of such a screening [42,43]. Accordingly, we stress that it is currently not justified to recommend a general screening for vitamin D deficiency by measuring 25(OH)D concentrations in the whole general population. Nevertheless, considering that certain groups of individuals or patients are particularly prone to vitamin D deficiency and/or may particularly benefit from vitamin D treatment, we suggest, in line with the Endocrine Society, that 25(OH)D measurements should be considered in these groups, as listed in Table 4 [14].

Total serum 25(OH)D concentration, i.e., the sum of 25(OH)D_3_ and 25(OH)D_2_, is the accepted marker for the assessment of vitamin D status, as it best reflects vitamin D supply by all different sources, i.e., endogenous vitamin D synthesis in the skin, diet, supplements, and mobilization from tissue stores. Previous reports on a relatively high inter-assay and inter-laboratory variability of 25(OH)D measurements underscore the need for assay standardization and laboratory quality assurance [44,45]. In patients with vitamin D deficiency and certain related health issues, e.g., bone diseases, it should be considered to measure additional laboratory parameters, including serum calcium, phosphate, alkaline phosphatase, parathyroid hormone (PTH), creatinine (to calculate the estimated glomerular filtration rate), and magnesium, as these laboratory markers may be useful to guide further diagnostics and treatment of these patients. Measurements of, e.g., serum calcium and creatinine are, however, also advised in patients with 25(OH)D concentrations above 100 ng/mL (250 nmol/L), as vitamin D oversupply/toxicity may lead to hypercalciuria, followed by hypercalcemia, potential acute kidney disease, and vascular calcification. Hypercalcemia does, however, usually not occur at 25(OH)D concentrations below 150 ng/mL (375 nmol/L) [46]. There are hardly any contraindications to correct vitamin D deficiency by vitamin D supplementation (e.g., kidney stones are *per se* no contraindication) except of rare conditions with an increased sensitivity to vitamin D treatment, such as inherited 24-hydroxylase-deficiency [47]. This is a rare genetic disorder in which catabolism of vitamin D metabolites is impaired, leading to hypercalcemia, low PTH concentrations, and relatively high serum 25(OH)D concentrations along with an increased risk of nephrolithiasis [47]. If such a disease is suspected, the measurement of 24,25-dihydroxyvitamin D, in a specialized laboratory, aids in the diagnosis as a high ratio of 25(OH)D to 24,25-dihydroxyvitamin D suggests this disease that is further confirmed by genetic analyses [47].

Classification of vitamin D status and its terminology, according to 25(OH)D concentration, remains a controversial issue in the scientific literature [7,16,21]. Being aware that it is an individual continuum from vitamin D deficiency to a sufficient and optimal vitamin D status, as well as to vitamin D toxicity, we suggest a classification system, as indicated in Table 4. It should be kept in mind that such a general classification of vitamin D status cannot take into account variations in the individual sensitivity to vitamin D effects that may be due to genetic polymorphisms, epigenetic or nutritional factors (e.g., magnesium status), as well as co-morbidities or medications [48,49,50,51,52].

**Table 4 nutrients-14-01483-t004:** Statement regarding screening of vitamin D deficiency in adults.

Consensus Statement	Consensus Voting Scale	Level of Agreement
Screening of vitamin D deficiency should be considered in the following patients/individuals or conditions: Osteoporosis; Osteomalacia; Musculoskeletal pain; Chronic kidney disease; Hepatic failure; Malabsorption syndromes (e.g., cystic fibrosis, inflammatory bowel diseases, bariatric surgery, radiation enteritis); Hyperparathyroidism; Chronic treatment with medications that influence vitamin D metabolism (e.g., antiseizure medications, glucocorticoids, AIDS-medications, antifungal agents, cholestyramine); Chronic autoimmune diseases (e.g., multiple sclerosis, rheumatoid arthritis); Pregnant and lactating women; Institutionalized or hospitalized patients; Older adults (>65 years) in general; Older adults with history of falls or nontraumatic fractures; Granuloma-forming disorders (e.g., sarcoidosis, tuberculosis, histoplasmosis, berylliosis, coccidiomycosis); Obesity (BMI ≥ 30kg/m^2^); dark skin pigmentation. 25(OH)D is recommended as a laboratory marker for the diagnosis of vitamin D deficiency. In patients with diagnosed vitamin D deficiency (25(OH)D < 20 ng/mL (<50 nmol/L)) and suspected related health issues, serum calcium, phosphate, alkaline phosphatase, parathyroid hormone (PTH), creatinine, and serum magnesium levels should be considered for evaluation; in particular in individuals with a 25(OH)D concentration of <10 ng/mL (<25 nmol/L). A 25(OH)D concentration of <20 ng/mL (<50 nmol/L) is considered as vitamin D deficiency A 25(OH)D concentration of ≥20 ng/mL (≥50 nmol/L) and <30 ng/mL (<75 nmol/L) is considered as vitamin D insufficiency A 25(OH)D concentration of 30–50 ng/mL (75–125 nmol/L) is considered as vitamin D sufficiency A 25(OH)D concentration of >50–60 ng/mL (125–150 nmol/L) is considered as safe but not as a target level A 25(OH)D concentration of >60–100 ng/mL (150–250 nmol/L) is considered as area of uncertainty with potential benefits or risks. A 25(OH)D concentration of >100 ng/mL (250 nmol/L) is considered as oversupply/vitamin D toxicity	9 (strongly agree)	50%
	8	20%
	7 (agree)	30%
	6	0%
	5 (neutral)	0%
	4	0%
	3 (disagree)	0%
	2	0%
	1 (strongly disagree)	0%

Overall agreement 100%, consensus endorsed		

### 3.3. Prevention of Vitamin D Deficiency in Adults

Most nutritional vitamin D guidelines conclude that vitamin D requirements are met for the vast majority (i.e., 97.5%) of the population when achieving a target 25(OH)D concentration of at least 20 ng/mL (50 nmol/L) [11,12]. Recommended dietary reference intakes for vitamin D usually range from 600 to 800 international units (IU) (40 IU are equal to 1 µg) per day and should ensure a sufficient vitamin D status under conditions of minimal-to-no sunlight exposure [11,12,13,16,53,54]. These vitamin D intake doses were calculated according to meta-regression analyses of so called “winter RCTs” to estimate the dose-response curve of vitamin D intakes and achieved serum 25(OH)D concentrations without relevant endogenous vitamin D synthesis in the skin [11]. It is a major limitation of most nutritional vitamin D guidelines that they performed meta-regression analyses based on aggregate data because such an approach does not adequately capture between person variability in the treatment response [53,54]. Using individual participant data instead of aggregate data for meta-regression analyses, as a superior methodological approach, results in significantly higher vitamin D intakes to achieve certain target 25(OH)D concentrations [53,54]. Individual participant data meta-regression analyses and single RCTs suggest that an overall vitamin D intake of about 1000 IU of vitamin D per day is required to maintain 25(OH)D concentrations of, at least, 20 ng/mL (50 nmol/L) in 97.5% of the population [54,55]. Therefore, we recommend a vitamin D supplement dose of at least 800 IU per day when targeting a sufficient vitamin D status, i.e., a 25(OH)D concentration of at least 20 ng/mL (50 nmol/L). We can, of course, improve and maintain vitamin D status by consuming natural or fortified food sources, but vitamin D intake by diet is usually in the range of about 100 to 200 IU per day in the general population [37,56].

In detail, we recommend a vitamin D supplementation dose of 800 to 2000 IU per day for adults who want to ensure a sufficient vitamin D status, with up to 4000 IU per day for certain groups, particularly for patients with obesity and malabsorption syndromes, as well as for individuals with a dark skin pigmentation (see Table 5). The relatively wide dose ranges for vitamin D account for various differences in the dose-response relationship for a given supplemental vitamin D dose and the achieved 25(OH)D concentration with higher dose requirements with increasing body weight and vice versa [57,58,59,60,61,62,63,64]. If a clinician is asked by a random individual which vitamin D dose is safe and very likely avoids vitamin D deficiency, a dose of 800 to 1000 IU per day should fulfill these criteria for the vast majority, even if individual characteristics, including the 25(OH)D status, is unknown. It should, however, also be noted that a few health authorities and experts consider a 25(OH)D concentration, of at least 10–12 ng/mL (25 to 30 nmol/L), as a reasonable treatment target that can be achieved by supplementation of 400 IU of vitamin D per day [11,13,16,53].

Daily, weekly, or monthly vitamin D supplementation, at equivalent doses, lead to similar increases in 25(OH)D serum concentrations, when measured after 2 to 3 months [65,66,67]. Adherence may be better with intermittent vitamin D dosing, but there are also concerns that high intermittent vitamin D doses may be less beneficial or might even be harmful in certain settings [67,68,69]. In view of the available evidence from clinical vitamin D trials and some pathophysiological considerations (e.g., altered vitamin D metabolism with high intermittent vitamin D doses), a daily vitamin D dosing schedule should rather be preferred, but when exceedingly high intermittent vitamin D doses are avoided, a weekly or monthly dosing schedule can also be applied [66,67]. The panel members could not reach a clear consensus on a clear cut-off for exceedingly high vitamin D doses, but single doses above about 50,000 IU of vitamin D should rather be avoided. Due to superior evidence regarding clinical benefits and dose-response, we rather prefer vitamin D3 (cholecalciferol) over vitamin D2 (ergocalciferol) for the prevention of vitamin D deficiency [51,70].

**Table 5 nutrients-14-01483-t005:** Statement regarding prevention of vitamin D deficiency in adults.

Consensus Statement	Consensus Voting Scale	Level of Agreement
In healthy adults without other risk factors, a supplementation of 800–2000 IU/day, for those who want to achieve a targeted/measured 25(OH)D concentration, should be considered during wintertime (mainly November-April) due to insufficient endogenous dermal vitamin D synthesis and depending on the body weight. Due to decreased skin synthesis in elderly (>65 years), a supplementation of 800–2000 IU/day is recommended throughout the year. In hospitalized/institutionalized individuals, a supplementation of 800–2000 IU/day is recommended throughout the year. Women planning a pregnancy should start or maintain the vitamin D supplementation as recommended for healthy adults without other risk factors (800–2000 IU/day). The vitamin D supplementation should be continued throughout pregnancy and lactation. In certain patients/individuals or conditions 2–3 times higher vitamin D dosages, without using vitamin D doses above the UL of 4000 IU/day, are recommended for prevention compared to healthy adults without other risk factors: Malabsorption (e.g., cystic fibrosis, inflammatory bowel diseases, bariatric surgery, radiation enteritis) Obesity (BMI ≥ 30 kg/m^2^) Dark skin pigmentation As vitamin D metabolites are stored in fat and other tissues and gradually released into the blood circulation, a daily or weekly or monthly supplementation regimen is equally effective and safe, if monthly doses are not exceedingly high, for the prevention of vitamin D deficiency. A tailored approach for vitamin D administration, involving the patients’ preferences of the supplementation regimen (daily, weekly, monthly) might enhance the adherence to preventive vitamin D supplementation. For the prevention of vitamin D deficiency, the supplementation of oral cholecalciferol (vitamin D3) is recommended.	9 (strongly agree)	30%
	8	20%
	7 (agree)	50%
	6	0%
	5 (neutral)	0%
	4	0%
	3 (disagree)	0%
	2	0%
	1 (strongly disagree)	0%

Overall agreement 100%, consensus endorsed		

### 3.4. Treatment of Vitamin D Deficiency in Adults

Individuals with a measured 25(OH)D concentration below 20 ng/mL (50 nmol/L) should be treated with vitamin D supplementation, because their vitamin D requirements may not be met [11,12]. There is controversy in the scientific literature whether 25(OH)D concentrations between 20 ng/mL (50 nmol/L) and <30 ng/mL (75 nmol/L) justify vitamin D supplementation [71,72]. The recommended dose range of 800 to 2000 IU per day is a reflection of various considerations underlying such treatment goals (see Table 6). When aiming for a minimum 25(OH)D concentration of at least 20 ng/mL (50 nmol/L), a daily vitamin D supplement dose of about 800 IU per day is sufficient for almost all individuals, even during the winter season, in Europe [54,55]. Data are less clear on which vitamin D doses are required to achieve a 25(OH)D concentration of ≥30 ng/mL (75 nmol/L) in almost all patients, but doses may be in the range of about 1500 to 2000 IU per day or even higher [14,21,26,27,28]. The classic rule of thumb that 100 IU of vitamin D per day increases serum 25(OH)D concentrations by about 1 ng/mL (2.5 nmol/L) seems to be a useful approximation, but several factors modulate the individual treatment response [73,74]. For example, increases in 25(OH)D are relatively high at low vitamin D supplement doses and low baseline 25(OH)D concentrations, whereas the dose-response curve flattens with higher vitamin D supplement doses and higher baseline 25(OH)D concentrations [73,74]. Evaluations of treatment success, by measurements of 25(OH)D, may be considered in certain patients, such as those with e.g., malabsorption or questionable adherence, but this should not be done earlier than 6 to 12 weeks after starting vitamin D supplementation, as this is about the time that it takes to reach a steady-state in serum 25(OH)D concentrations [2]. Although there is, of course, a seasonal variation in serum 25(OH)D concentrations, usually with higher levels during summertime, as a consequence of endogenous vitamin D synthesis in the skin, we do, in general, recommend continuous and, usually, fixed doses of vitamin D supplementation throughout the year. The decrease in serum 25(OH)D during winter season is, in many patients, significant, but it is less than could be expected by the half-life of serum 25(OH)D concentrations of about 2 to 3 weeks because of a mobilization of vitamin D and its metabolites from various tissue stores (e.g., adipose tissue and muscle) [75].

If a rapid correction of vitamin D deficiency is clinically indicated, a regimen with a higher initial vitamin D dose, i.e., 6000 IU per day, and in certain cases, even up to 10,000 IU per day, followed by a maintenance dose with 800 to 2000 IU per day is recommended (Table 6). Such doses of 6000 IU, or even up to 10,000 IU, per day for several weeks are usually safe and ensure a more rapid correction of vitamin D deficiency compared to lower doses [14,76,77]. Daily vitamin D doses are generally preferred over intermittent dosing schedules [67]. The clinical indications for a rapid correction of vitamin D are, beyond osteomalacia, not clearly defined but may, according to our opinion, involve conditions such as extremely low 25(OH)D concentrations, osteoporosis patients with a very high fracture risk, patients with secondary hyperparathyroidism, and/or reduced serum calcium concentrations.

Regarding treatment of vitamin D deficiency and its prevention, we want to emphasize that promoting a healthy lifestyle by preventing or reducing obesity, regular physical activity with moderate (cautious) sunlight exposure, and a healthy balanced diet are also effective measures to improve both vitamin D status and overall health. Promoting such lifestyle measures is, of course, also highly recommended, and it should accompany any vitamin D treatment.

As for the prevention of vitamin D deficiency, we recommend vitamin D3 (cholecalciferol) over vitamin D2 (ergocalciferol) for its treatment. Although parenteral, particularly intramuscular, vitamin D treatment can be considered in patients with malabsorption, e.g., inflammatory bowel disease, we primarily suggest to increase the oral vitamin D dose in such settings [78]. Clinicians have to consider that patients with inflammatory bowel disease frequently require higher vitamin D doses, but with daily oral vitamin D supplementation of about 5000 to 10,000 IU, even these patients usually achieve their 25(OH)D target concentrations [78]. If intermittent intramuscular vitamin D injections (e.g., 100,000 IU all three months) are used, the doses are roughly similar and slightly more efficient for intramuscular compared to oral doses, in terms of raising serum 25(OH)D concentrations, but the increase in serum 25(OH)D is slower for intramuscular versus oral vitamin D supplementation [78,79,80,81].

Some experts argue that calcifediol (=25(OH)D3, calcidiol) may also be used to correct vitamin D deficiency in certain conditions. The use of calcifediol seems to be more justified in obese people, people with malabsorption syndromes, people with liver disease, patients suffering from chronic kidney disease (stage 3 or 4), and those in all conditions where rapid correction of vitamin D deficiency is required [82,83,84]. Furthermore, calcifediol use may also be beneficial in patients taking medications that disrupt the hepatic cytochrome P-450 enzyme system, including those taking glucocorticoids, anticonvulsants, anticancer drugs, or antiretroviral drugs [82,83,84,85]. The increase in serum 25(OH)D is markedly reduced in patients with obesity (high BMI) and in patients with malabsorption syndromes treated with cholecalciferol, but with calcifediol, the 25(OH)D increase is not significantly different according to BMI or according to the presence, or absence, of malabsorption syndromes. Moreover, the increase in serum 25(OH)D is faster, and the dose-response curve is more linear with the use of calcifediol versus vitamin D3, and when stopping treatment, the decline in 25(OH)D concentration is faster after calcifediol compared to vitamin D3 [75,82,83,84]. While accumulating evidence suggests that calcifediol may be an attractive alternative to “native” vitamin D, due to the lack of experience with this molecule in Central and Eastern European countries, at this stage, we continue to recommend vitamin D3 (cholecalciferol) [75]. Cholecalciferol and calcifediol appear, so far, as equal molecules in the fight with vitamin D deficiency. However, RCT data are still missing on the superior benefit of calcifediol versus vitamin D, with reference to hard clinical outcomes, but more data on this topic may be available in the future [75,84,85].

Calcitriol (=1,25(OH)2D) and its analogues are used at much lower doses compared to vitamin D3, have a relatively high risk of hypercalcemia and a relatively narrow therapeutic window, and are not recommended for the treatment of common vitamin D deficiency [64]. Therefore, active vitamin D treatment is only indicated in certain diseases, such as chronic hypoparathyroidism, chronic kidney disease, or mineral and bone disorders (CKD-MBD) [64,86].

**Table 6 nutrients-14-01483-t006:** Statement regarding treatment of vitamin D deficiency in adults.

Consensus Statement	Consensus Voting Scale	Level of Agreement
It is recommended to initiate a vitamin D deficiency treatment at a 25(OH)D concentration of <20 ng/mL (<50 nmol/L). At a concentration of <30 ng/mL (<75 nmol/L) a treatment may be considered. Individuals with diagnosed vitamin D deficiency can be initially treated with higher vitamin D dosages compared to the preventive dosages recommended for the general population, if a rapid correction of the 25(OH)D concentration is clinically indicated. As initial dose for the treatment of vitamin D deficiency in patients without other risk factors, a dosage of 6000 IU, equivalent to a daily dosage, is recommended. In certain individuals or conditions, higher vitamin D dosages, up to 10,000 IU, equivalent to a daily dosage, are recommended for treatment compared to healthy adults without other risk factors (Endocrine Society recommendation) [14]: -Malabsorption (e.g., cystic fibrosis, inflammatory bowel diseases, bariatric surgery, radiation enteritis) -Chronic treatment of medications that influence vitamin D metabolism (e.g., antiseizure medications, glucocorticoids, AIDS-medications, antifungal agents, cholestyramine) -Obesity (BMI ≥ 30 kg/m^2^) A treatment duration of 4–12 weeks is recommended, depending on the severity of vitamin D deficiency. A tailored approach for vitamin D administration, involving the patients’ preferences of the treatment regimen (daily, weekly, monthly) might enhance the adherence to the therapy. As soon as a 25(OH)D concentration of 30–50 ng/mL (75–125 nmol/L) is achieved, a maintenance dose of 800–2000 IU/day is recommended, that can also be used as an initial treatment dose if there is no requirement for a rapid correction of vitamin D deficiency. Approx. 6–12 weeks after start of the treatment, the effectiveness may be evaluated by measurement of the 25(OH)D concentration particularly in certain risk groups with e.g., malabsorption syndrome. For the treatment of vitamin D deficiency in adults, oral cholecalciferol (vitamin D3) is preferred. Calcifediol may be used instead of vitamin D in certain conditions, including obesity or malabsorption. Calcitriol and active vitamin D analogues may be considered in special patient groups. In certain risk groups (e.g., patients with severe malabsorption), parenteral vitamin D treatment can be considered.	9 (strongly agree)	60%
	8	10%
	7 (agree)	30%
	6	0%
	5 (neutral)	0%
	4	0%
	3 (disagree)	0%
	2	0%
	1 (strongly disagree)	0%

Overall agreement 100%, consensus endorsed		

### 3.5. Vitamin D in Musculoskeletal Disorders

While older meta-analyses of vitamin D RCTs showed a significant reduction in fractures by vitamin D supplementation at a daily dose of about 800 to 2000 IU per day, these data have recently been challenged by updated meta-analyses, suggesting no significant effect on fractures and falls [5,87,88]. Nevertheless, major osteoporosis guidelines recommend vitamin D treatment in osteoporosis patients, and some studies indicate that sufficient 25(OH)D concentrations are required for optimal bisphosphonate treatment efficacy [51,89]. While it is beyond the scope of our work to release detailed recommendations regarding calcium supplementation in osteoporosis patients, we wish to point out that a recent RCT with the bisphosphonate zoledronate showed excellent anti-fracture effects in patients using pure vitamin D supplementation without additional calcium supplements but consuming 1g of calcium daily by a usual diet [90]. These data may suggest that, even in osteoporosis patients, vitamin D treatment without additional calcium supplements may be sufficient in the case of adequate dietary calcium intake, but this issue is still not clarified in the scientific community, since it is challenging to disentangle the effects of vitamin D and calcium with regards to bone health [5,88,90,91,92]. Of note, increasing calcium and protein intake by milk, yoghurt, and cheese significantly reduces the risk of falls and fractures in aged care residents [93]. Calcium supplementation is, however, generally indicated in patients with osteoporosis and an insufficient dietary calcium intake. In this context, it should be stressed that most RCTs on osteoporosis drugs were conducted under the conditions of combined calcium plus vitamin D supplementation, thus supporting the use of calcium supplementation, in addition to vitamin D for osteoporosis treatment. Being aware of the uncertainties regarding vitamin D, in the context of osteoporosis, we nevertheless strongly argue to ensure a sufficient vitamin D status in patients with an increased risk of fractures and falls (Table 7). With reference to falls, some studies suggest that high-dose intermittent vitamin D supplementation may even be detrimental so that we prefer daily doses at the lower end of the dosing range of 800 to 2000 IU per day, in this setting [67,85,94]. Therefore, vitamin D overdosing must particularly be avoided in older and severely ill patients [94,95].

**Table 7 nutrients-14-01483-t007:** Statement regarding vitamin D in musculoskeletal disorders.

Consensus Statement	Consensus Voting Scale	Level of Agreement
In osteoporosis patients, a supplementation of 800–2000 IU/day, with oral cholecalciferol (vitamin D3) is recommended in combination with calcium, if indicated. Vitamin D deficiency may impair the response to the osteoporosis treatment, thus screening of the 25(OH)D level and correction of vitamin D deficiency before starting the osteoporosis treatment with antiresorptive medications is recommended. In patients with an increased risk of falls or fractures, a supplementation of 800–2000 IU/day is recommended.	9 (strongly agree)	30%
	8	10%
	7 (agree)	60%
	6	0%
	5 (neutral)	0%
	4	0%
	3 (disagree)	0%
	2	0%
	1 (strongly disagree)	0%
Overall agreement 100%, consensus endorsed		

### 3.6. Extra-Skeletal Actions of Vitamin D in Adults

The expression of the VDR and of vitamin D-metabolizing enzymes in almost all human tissues and cells suggests a widespread role of vitamin D for overall human health [1,10]. In line with this, numerous epidemiological studies documented that low 25(OH)D concentrations are associated with an increased risk of all-cause mortality and major acute and chronic diseases, such as cancer, cardiovascular and autoimmune diseases, as well as infections (see Table 8) [1,3,4,6,8,92,96,97,98,99,100]. Confounding and reverse causation may, however, explain parts of these associations so that cause and effect relationships are not yet fully established for several of the above mentioned diseases [101].

Meta-analyses of vitamin D RCTs suggest that vitamin D supplementation may reduce the incidence of acute respiratory infections, cancer mortality, as well as asthma and chronic obstructive pulmonary disease (COPD) exacerbations [5,96,102,103,104,105,106,107,108]. Considering the totality of evidence regarding vitamin D and a variety of extra-skeletal diseases, we are of the opinion that it is justified to consider screening and preventive vitamin D supplementation in certain populations at risk (see Table 8). The general dilemma with potential extra-skeletal health benefits of vitamin D in the context of vitamin D guidelines is that vitamin D requirements for skeletal health, such as for the prevention of rickets and osteomalacia, may be met at lower 25(OH)D concentrations than the requirements for certain extra-skeletal health benefits [1,14]. Of note, recent large vitamin D RCTs failed to document significant benefits regarding their primary outcomes, such as mortality, cancer, or cardiovascular diseases, but these trials enrolled populations that were, by the vast majority, not vitamin D deficient [109,110].

Numerous studies have been published on vitamin D, in the context of the Coronavirus Disease 2019 (COVID-19) pandemic, caused by the severe acute respiratory syndrome coronavirus 2 (SARS-CoV-2) [111,112,113]. While there is accumulating evidence suggesting potential benefits of vitamin D or calcifediol, for the prevention and treatment of COVID-19, this hypothesis still requires confirmation in large clinical trials [111,112,113]. Fortunately, due to the efficacy of vaccines and natural immunity of individuals who recovered from SARS-CoV-2 infections, the COVID-19 pandemic has already been significantly mitigated in early 2022, with expected further improvements in the near future [114,115,116].

**Table 8 nutrients-14-01483-t008:** Statement regarding extra-skeletal actions of vitamin D in adults.

Consensus Statement	Consensus Voting Scale	Level of Agreement
Results from observational studies consider a low 25(OH)D concentration as a potential risk marker for several diseases such as cancer incidence and mortality, cardiovascular diseases, diabetes mellitus and its comorbidities, chronic autoimmune diseases, metabolic syndrome, acute respiratory tract infections, neurological diseases and total mortality. Results from meta-analyses of RCTs suggest that beyond musculoskeletal-effects, vitamin D supplementation may have beneficial extra-skeletal effects regarding acute respiratory tract infections and cancer mortality. In patients with or at risk of different types of cancer, certain cardiovascular diseases, diabetes mellitus and its comorbidities, chronic autoimmune diseases, certain neurological diseases and recurrent acute respiratory tract infections, a screening of vitamin D deficiency should be considered, and preventive vitamin D supplementation may be considered.	9 (strongly agree)	60%
	8	10%
	7 (agree)	30%
	6	0%
	5 (neutral)	0%
	4	0%
	3 (disagree)	0%
	2	0%
	1 (strongly disagree)	0%
Overall agreement 100%, consensus endorsed		

### 3.7. Development of a Vitamin D Deficiency Screening and Treatment Algorithm

Based on our consensus statements, we developed an algorithm for the prevention and treatment of vitamin D deficiency that was also subject to a Delphi voting round, with the following point scale results: 9 points (20% of panel members), 8 points (30%), and 7 points (50%) (see Figure 1).

## 4. Strengths and Limitations

As a potential limitation, the financial and logistic support by an industry sponsor (Wörwag Pharma) must be acknowledged. Despite no involvement of the sponsor in the scientific group discussions and consensus reaching processes, we cannot totally exclude some sort of funding bias [17,18]. Moreover, this consensus document was not informed by an a priori structured and pre-registered systematic review of the evidence. A definite strength of our work is the a priori structured process for developing this document, based on the Delphi method for reaching consensus. The number of 10 participants for using the Delphi voting rounds may be considered as too low, but such a group size is not uncommon and, thus, is generally accepted in the scientific literature on this methodology [35]. The involvement of several colleagues from Eastern Europe, who are usually underrepresented in European expert groups, and the well-balanced gender distribution of this panel, may also be regarded as a strength of our work.

## 5. Conclusions

This consensus statement covers various statements with relevance for the clinical practice in the prevention, diagnosis, and treatment of vitamin D deficiency. We highlighted the relevance of vitamin D for public health and provided guidance regarding this issue by considering the totality of the available scientific evidence, including our personal experience and opinions.

We consider that our work adds to the existing literature by providing a useful and evidence-based guidance for clinicians and health-care workers regarding several relevant and partially controversial topics for the practical management, with reference to vitamin D. In addition, we also addressed various issues with relevance for public health authorities and individuals from the general population that will, hopefully, help to reduce the global health burden of vitamin D deficiency.

## Figures and Tables

**Figure 1 nutrients-14-01483-f001:**
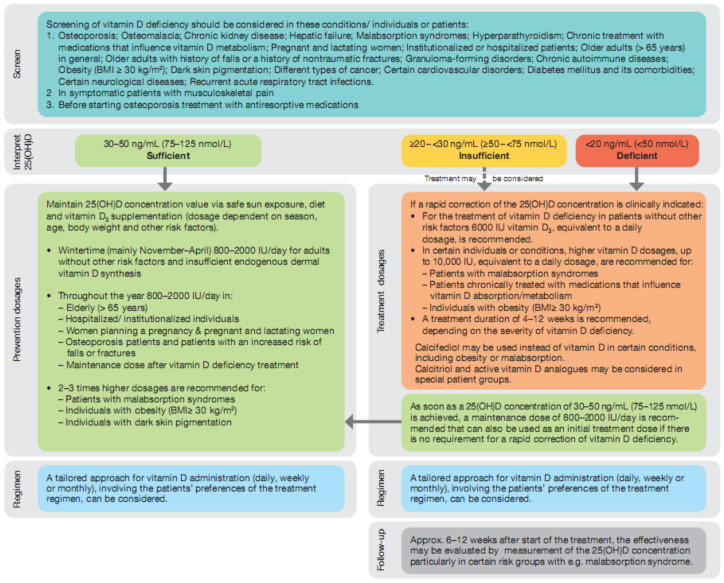
Algorithm for vitamin D deficiency screening and treatment.

**Table 1 nutrients-14-01483-t001:** Selected guideline recommendations for prevention of vitamin D deficiency in adults with a focus on Central and Eastern European countries, published since 2010.

Authority and/or Country or Region (Year)	Target Population	Age (Years)	Oral Vitamin D (IU)	Reference
Endocrine Society (2011) USA	General population	19–70	600–2000/day	Holick et al. [14]
		>70	800–2000/day	
	Pregnant and lactating women		600–2000/day	
	Obese individuals/Patients on anticonvulsants, glucocorticoids, antifungals, AIDS medications		2–3 times more	
DACH (2012) Germany/Austria/Switzerland	General population	>18	800/day	DGE [23]
EVIDAS (2013) Central Europe	General population	>18	800–2000/day	Płudowski et al. [21]
	Obese individuals and elderly		1600–4000/day	
	Prevention of pregnancy and fetal development complications	>16	1500–2000/day	
	Night workers and dark skin pigmentation		1000–2000/day	
EFSA (2016) Europe	General population	>18	600/day	EFSA [24]
Russia (2016)	General population	>18	800–1000/day	Pigarova et al. [25]
	Pregnant women		800–2000/day	
Poland (2018)	General population	19–75	800–2000/day	Rusińska, Płudowski et al. [26]
	Obese individuals	19–75	1600–4000/day	
	General population	>75	2000–4000/day	
	Obese individuals	>75	4000–8000/day	
	Pregnant and lactating women		2000/day	
Belarus (2013)	General population	>18	800–2000/day	Rudenko [27]
Hungary (2012)	General population	>18	1500–2000/day	Takács et al. [22]
	Pregnant and lactating women		1500–2000/day	
Bulgaria (2019)	General population	>19	600–2000/day	Borisova et al. [28]
	Pregnant and lactating women		600–2000/day	
	Patients on anticonvulsants, glucocorticoids, antifungals		2–3 times more	
Slovakia (2018)	Postmenopausal osteoporosis patients	>50	800–1000/day	Payer et al. [29]

**Table 2 nutrients-14-01483-t002:** Selected guideline recommendations for treatment of vitamin D deficiency in adults with a focus on Central and Eastern European countries, published since 2010.

Authority and/or Country or Region (Year)	Target Population	Oral Vitamin D for Treatment (IU)	Treatment Duration	25(OH)D Target Concentration nmol/L (ng/mL)	Oral Vitamin D for Maintenance (IU)	Reference
Endocrine Society (2011) USA	General population	50,000/week or 6000/day	8 weeks	75 (30)	1500–2000/day	Holick et al. [14]
	Obese individuals/Patients on anticonvulsants, glucocorticoids, antifungals, AIDS medications	2–3 times more; at least 6000–10,000/day			3000–6000/day	
EVIDAS (2013) Central Europe	General population	50,000/week or 7000–10,000/day	4–12 weeks	75–125 (30–50)	a maintenance dose may be instituted	Płudowski et al. [21]
Italy (2018)	General population	50,000/week or 5000/day	8 weeks	>75 (>30)	50,000 IU twice per month or 1500–2000 IU/day	Cesareo et al. [30]
Russia (2016)	General population	25(OH)D < 50 nmol/L (<20 ng/mL):		>75 (>30)	1000–2000/day or 6000–14,000/week	Pigarova et al. [25]
		50,000/week or	8 weeks			
		200,000/month or	2 months			
		150,000/month or	3 months			
		6000–8000/day	8 weeks			
		25(OH)D < 75 nmol/L (30 ng/mL):				
		50,000/week or	4 weeks			
		200,000 or	single dose			
		150,000 or	single dose			
		6000–8000/day	4 weeks			
Poland (2018)	General population	6000/day	12 weeks or until a 25(OH)D concentration of 75 nmol/L (30 ng/mL) is reached	>75–125 (>30–50)	maintenance dose i.e., a prophylactic dose recommended for the general population (see Table 1)	Rusińska, Płudowski et al. [26]
Belarus (2013)	General population	25(OH)D < 25 nmol/L (<10 ng/mL): 2000 to 10,000/day	4–12 weeks	75–200 (30–80)	800–2000 IU/day	Rudenko [27]
		25(OH)D 25–50 nmol/L (10–20 ng/mL): 800 to 4000/day	1 year			
Hungary (2012)	General population	50,000/week or	4–8 weeks	75 (30)	1500–2000/day	Takács et al. [22]
		30,000/week or	6–12 weeks			
		2000/day	12 weeks			
Bulgaria (2019)	General population	To maintain bone health: 1000–2000/day	-	50 (20)	maintenance dose i.e., a prophylactic dose recommended for the general population (see Table 1)	Borisova et al. [28]
		For extra–skeletal effects: 2000–4000/day	-	75–110 (30–44)		

## Data Availability

Not applicable.

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
