# Peer review of "Clinical Practice in the Prevention, Diagnosis and Treatment of Vitamin D Deficiency: A Central and Eastern European Expert Consensus Statement"

_nutrients, 2022, doi:10.3390/nu14071483_

Round 1

Reviewer 1 Report

This paper is a consensus statement in central and eastern Europe, and its content itself should be correct. I am just wondering about voting process, because one "agree" seems to include many questions in each table.  Author should clarify if they allowed "agreement in part and disagreement in part" in voting process. 

Author Response

Response: We thank the Reviewer 2 for this comment and we now clarify in the text that our consensus statements were grouped as listed in Tables 3 to 8 and Figure 1, so that each voting round was performed for the whole content of each Table or Figure, respectively. Group discussions before each voting round were however used to fine tune the statements according to the opinions and recommendations of the panel members. Therefore, each single part of any statement was subject to a group discussion. We hope that it is now clear how we structured this process and hope that the reviewer agrees with this.

Reviewer 2 Report

The authors of this expert consensus statement used the Delphi method to come to consensus recommendations regarding prevention, diagnosis and treatment of vitamin D deficiency. I consider myself also an expert in this field and would not have answered the questions in the Delphi process differently. Nevertheless, I have major concerns about the methodology that I think the authors cannot change anymore.

  • 10 experts is a too small number in my view. This means a minimum of 8 from 10 persons make a consensus statement happen. If you have by chance 3 experts with quite a diverging opinion from the other 7, almost nothing could be agreed on. On the other hand, if a strong minority opinion that e.g. 30% of experts in the field truly have, is not represented by at least 3 of the 10 experts chosen, it would not have found its way in the consensus statement. I would have had much more confidence in the expert consensus if it was based on at least 50 experts.
  • The selection of experts is not very transparent. Based on what qualifications were the experts chosen? For transparency, the names and qualifications of the experts who took part in the Delphi process should be published. In addition, their potential conflicts of interest should also be published.

The biggest concern is the funding by Wörwag Pharma. All authors received honoraria by Wörwag Pharma and also the meetings were organized by the pharma company. Wörwag Pharma sells Vitagamma – a high-dose vitamin D drug. Although the authors stated that they were not influenced by the funder, the reality is that such funding is always unconsciously playing a role in scientific work. In my opinion, it is a crossed red line when expert consensus statements are being sponsored by the pharmaceutical industry. Also the intended audience of the article – physicians in clinical care – will have no trust in this expert consensus statement when they read about the funding. Doing such a Delphi process with a few experts is not expensive and does not require external funding. I think it should be possible to do it with institutional funding from universities.

Author Response

Response: We thank the Reviewer 1 for the general comments above, in particular for acknowledging an agreement with the statements. We take the comment on the methodology seriously and agree that it is important to address this issue as outlined in our specific answers below.

  • 10 experts is a too small number in my view. This means a minimum of 8 from 10 persons make a consensus statement happen. If you have by chance 3 experts with quite a diverging opinion from the other 7, almost nothing could be agreed on. On the other hand, if a strong minority opinion that e.g. 30% of experts in the field truly have, is not represented by at least 3 of the 10 experts chosen, it would not have found its way in the consensus statement. I would have had much more confidence in the expert consensus if it was based on at least 50 experts.

Answer: We agree that 10 panel members may be considered too few for a Delphi round, as noted by the reviewer, and we now acknowledge this as a limitation of our work. It is, however, not uncommon to have such a number of participants as in the already cited reference by Diamond et al. (J Clin Epidemiol, 2014;67:401-409) 14% of the studies using the Delphi method had 10 participants or even less in the final voting round. Therefore, we included the following sentence in our manuscript and hope that the reviewer agrees with this: “The number of 10 participants for using the Delphi voting rounds may be considered as too low but such a group size is not uncommon and thus generally accepted in the scientific literature on this methodology.”  

  • The selection of experts is not very transparent. Based on what qualifications were the experts chosen? For transparency, the names and qualifications of the experts who took part in the Delphi process should be published. In addition, their potential conflicts of interest should also be published.

Answer: We agree that the selection process of the experts for the Delphi method is critical for our work. We have already outlined that the first and senior authors of this manuscript (i.e. Pawel Pludowski and Stefan Pilz) selected the other experts for this work with the aim to include respective experts from Central and Eastern European countries. The 10 authors of this paper are the 10 experts who participated in the Delphi voting rounds. We have now clearly mentioned this in the manuscript and apologize for being unclear with this in our initial submission. Regarding conflicts of interest we have already declared them in the Conflicts of Interest statement at the end of the manuscript. For details on this very important issue see our answer to the comment below.    

The biggest concern is the funding by Wörwag Pharma. All authors received honoraria by Wörwag Pharma and also the meetings were organized by the pharma company. Wörwag Pharma sells Vitagamma – a high-dose vitamin D drug. Although the authors stated that they were not influenced by the funder, the reality is that such funding is always unconsciously playing a role in scientific work. In my opinion, it is a crossed red line when expert consensus statements are being sponsored by the pharmaceutical industry. Also the intended audience of the article – physicians in clinical care – will have no trust in this expert consensus statement when they read about the funding. Doing such a Delphi process with a few experts is not expensive and does not require external funding. I think it should be possible to do it with institutional funding from universities.

Answer: We totally understand this comment and agree with the reviewer that this is a relevant concern. We therefore, already mentioned and discussed this as a limitation of our work in the submitted manuscript version. We did this in a very self-critical and open way by even including this limitation in the manuscript text and not only in the acknowledgement section and Conflicts of Interest statement. We do hope that this openness will not be punished and that the readership gives us credit for this transparency rather than considering this as a red line. It is for us important to note that many other guidelines or consensus working groups indicate that they are not funded by industry and thus claim to be independent. However, we clearly know from the literature that this is not the case and it is thus debatable whether this is so much more scientifically sound than our approach (see e.g. Moynihan R, et al. BMJ 2020;369:m1505; Nusrat S, et al. JAMA Netw Open, 2018;1(8):e186343; Checketts, JX et al. JAMA Dermatol, 2017;153(12):1229-1235; Mitchell, AP et al. Oncologist, 2021;26(9):771-778). It is common practice that medical associations are sponsored by the industry as well as the authors of guidelines, but when a guideline conference is organized it is just indicated that the medical association is sponsoring this without any industry involvement. This criticism of transparency of industry sponsorship does, of course, not anyhow mean that industry sponsorship does not have a potential impact on our work. We really take this limitation very seriously because otherwise we would not have stressed this with so much emphasis in our initial submission. On the other hand, we do not agree that our universities or personal sponsorship could have made such a group effort possible. Therefore, we hope that the reviewer agrees that with this very transparent and open way of noting the limitation by industry sponsorship of our work, and that our efforts are nevertheless worth publishing, in particular as the readership receives all the relevant information on how this work has been conducted and what are the potential limitations.

Round 2

Reviewer 2 Report

Thank you very much for your answers. What turned my view on this manuscript was the fact that other medical guidelines were also not conducted without industry funding and that you were really open about this from the start. Thus, as you said, we can leave it up to the readers, how much they want to trust this expert consensus.